# Benefit of sequential bilateral cochlear implantation in children between 5 to 18 years old: A prospective cohort study

W. J. Kleijbergen[1], M. Sparreboom[2], E. A. M. Mylanus[2], G. de Koning[1], H. W. Helleman[1], P. P. B. M. Boermans[3], J. H. M. Frijns[3,4], J. L. Vroegop[5], M. P. van der Schroeff[5], E. E. J. Gelders[6], E. L. J. George[6], M. J. W. Lammers[1,7], W. Grolman[7,8], I. Stegeman[1,7], A. L. Smit[1,7]*

1 Department of Otorhinolaryngology, Head and Neck Surgery, University Medical Center Utrecht, Utrecht, The Netherlands, 2 Department of Otorhinolaryngology, Head and Neck Surgery, Hearing and Implants, Donders Institute for Brain, Cognition and Behaviour, Radboud University Medical Centre, Nijmegen, The Netherlands, 3 Department of Otorhinolaryngology and Head and Neck Surgery, Leiden University Medical Center, Leiden University, Leiden, The Netherlands, 4 Leiden Institute for Brain and Cognition, Leiden, The Netherlands, 5 Department of Otorhinolaryngology and Head and Neck Surgery, Erasmus Medical Center, Erasmus University, Rotterdam, The Netherlands, 6 Department of Otorhinolaryngology and Head and Neck Surgery, Maastricht University Medical Center, Maastricht University, Maastricht, The Netherlands, 7 Brain Center Rudolf Magnus, University Medical Center Utrecht, Utrecht, The Netherlands, 8 Causse Ear Clinic, Tertiary Ear Referral Center, Colombiers, France

* A.L.Smit-9@umcutrecht.nl

## Abstract

### Objective

To determine the benefit of sequential cochlear implantation after a long inter-implantation interval in children with bilateral deafness receiving their second implant between 5 and 18 years of age.

### Study design

Prospective cohort-study.

### Setting

Tertiary multicenter.

### Patients

85 children with bilateral deafness and unilateral implantation receiving a contralateral cochlear implant at the age of 5 to 18 years.

### Method

The primary outcomes were speech recognition in quiet and noise (CVC) scores. The secondary outcomes were language outcomes and subjective hearing abilities, all measured before and 12 months after sequential bilateral cochlear implantation. Medians of the paired

**Data Availability Statement:** Data cannot be shared publicly because of the non-identifiability of

the data is not guaranteed by a de-identified data set based on the specific variables included of the persons in the data set. By the multicenter design we were relying on data transfer agreements set previously between participating centers to aggregate the data used for this study. In this agreement it was stated that 'UMC Utrecht will have the right to disclose non-individually identifiable information regarding the Data in a summary form that aggregates more than one individual's clinical information for scientific journal publication'. Based on this legal issues, at this stage we could only allow individual data availability upon request by amendments of the data transfer agreements which have to be made upon request. Though we do allow aggregated data to be available under current legal conditions. Data are available from the dHS research office UMCU (contact via e-mail: DHS_onderzoeksbureau@umcutrecht.nl) for researchers who meet the criteria for access to confidential data. Address correspondence to Department of Otorhinolaryngology, Head and Neck Surgery, University Medical Center Utrecht, PO Box 85500, 3508 GA Utrecht, The Netherlands.

**Funding:** The authors received no specific funding for this work.

**Competing interests:** The authors have declared that no competing interests exist.

**Abbreviations:** AB, Advanced Bionics; BICI, bilateral cochlear implantation; CELF, Clinical Evaluation of Language Fundamentals; CI, cochlear implant; CI2, second cochlear implant; CI-center, cochlear implantation center; CMV, cytomegalovirus; CVC, consonant-vowel-consonant; dB, decibel; DFN, deafness; EVA, Enlarged Vestibular Aqueduct; EMC, Erasmus Medical Center; HA, hearing aid; Hz, hertz; kHz, kilohertz; LUMC, Leiden University Medical Center; MUMC, Maastricht University Medical Center; NVA, Nederlandse Vereniging van Audiologie (Dutch Audiology Society); PPVT, Peabody Picture Vocabulary Test; Radboud MC, Radboud University Nijmegen Medical Center; SD, standard deviation; SNHL, sensorineural hearing loss; SNR, signal to noise ratio; SPL, Sound Pressure Level; SSQ, speech, spatial, and qualities of hearing scale; UCI, unilateral cochlear implantation; UMCU, University Medical Center of Utrecht.

data were compared using the Wilcoxon signed-rank test. Univariable linear regression analyses was used to analyze associations between variables and performance outcomes.

## Results

A significant benefit was found for speech recognition in quiet (96% [89–98] vs 91% [85–96]; $p < 0.01$) and noise (65% [57–75] vs 54% [47–71]; $p = 0.01$) in the bilateral CI condition compared to unilateral (n = 75, excluded 10 non-users). No benefit was seen for language outcomes. The subjective sound quality score was statistically significant higher in bilateral compared to the unilateral CI condition. Pre-operative residual hearing level in the ear of the second implant, the inter-implant interval and age at time of second implantation was not significantly associated with performance scores.

## Conclusion

After 12 months of use, sequential bilateral cochlear implantation showed improved speech perception in quiet and noise and improved subjective sound quality outcomes in children despite a great inter-implantation interval (median of 8 years [range 1–16 years]).

## Introduction

In recent years, literature demonstrating that bilateral cochlear implantation in children results in superior hearing outcomes compared to unilateral implantation steadily increases [1]. Despite the fact that unilateral cochlear implantation (UCI) gives generally rise to good speech perception performance in quiet conditions, bilaterally implanted children perform better in noisy environments (i.e. classrooms) and more challenging hearing conditions [2–4]. Moreover, children with bilateral cochlear implantation (BICI) achieve better sound localization [5,6], speech and language development [7–10], and self-rating outcomes [11] when the second implant is provided simultaneously (within a single surgery) or sequentially within a short inter-implant interval [12,13]. As a result, in numerous countries bilateral cochlear implantation has become the standard of care as treatment for young children with bilateral severe to profound sensorineural hearing loss (SNHL). In the Netherlands from 2013 onwards, bilateral cochlear implants became available for all children with bilateral sensorineural hearing loss up to the age of five years. Shortly after this milestone, a conditional reimbursement was granted for all children aged five to eighteen years for a second cochlear implant. For this, outcome evaluation on a national level was requested by the Dutch health care institute as a necessity for this refund.

The timing of auditory stimulation in relation to the process of cortical development is considered as a major factor of importance to explain variations in outcomes of children receiving their second cochlear implant in later childhood [14]. If the stimulation by the second CI is out of the sensitive period, which is described to be approximately around 7 years of age, this could limit functional outcome [15]. Therefore, simultaneous BICI or sequential implantation within a short interval is advised for children with bilateral deafness [16,17]. Nonetheless, there is still a cohort of children with bilateral hearing loss who were implanted unilaterally early in life, comprised of children with prelingual deafness and children with progressive hearing loss. These children either experience unilateral stimulation by the cochlear implant, or bimodal input by wearing a contralateral hearing aid before being implanted with their

second implant. Children from this cohort with demonstrated effectiveness of the first implant are thought to qualify for a second (contralateral) cochlear implant. This is substantiated by earlier evidence showing the advantage of sequential bilateral implantation for speech recognition in quiet and noise and the receptive vocabulary [18–19]. In the long term, the improvement in the more basic language skills, like receptive vocabulary, may also lead to an improvement in more complex linguistic skills.

In case of sequential implantation, it is demonstrated that not only a short inter-implant interval but also younger age, better residual hearing and bimodal stimulation favors functional outcomes [20,21]. So far, most studies analyzing outcome of sequentially implanted children included populations with a mean age between 5 to 10 years at the time of the second cochlear implantation [3,17,20,21]. Only a few studies reported outcomes even in cases receiving their second implant during adolescence [14,22]. For this group of children, receiving their second cochlear implant many years after their first implant, the association between these factors and hearing performance is unclear. Following the conditional reimbursement for the second implant for the children aged five to eighteen years in the Netherlands, a Dutch national multicenter prospective trial was initiated to assess the benefit of the second cochlear implant in terms of speech perception, subjective performance of hearing abilities and speech and language development after one year.

## Materials and methods

### Study objective, design and participants

In this study we aim to assess hearing outcomes in children receiving their second cochlear implant in the age of 5–18 years after being previously unilaterally implanted at a younger age. Secondly, we aim to analyze the influence of patient related characteristics on speech reception and speech and language development scores with their second implant.

The data in this multicenter prospective cohort study were collected from January 2014 to December 2016, in which five cochlear implantation centers (CI-centers) in the Netherlands participated: Radboud University Medical Center Nijmegen (Radboudumc), the University Medical Center of Utrecht (UMCU), Leiden University Medical Center (LUMC), Erasmus Medical Center (EMC) and Maastricht University Medical Center (MUMC). Patients eligible to participate in this study were previously unilaterally implanted with a cochlear implant, where some of these children wore a hearing aid in addition (bimodal stimulation) to their cochlear implant. To be eligible to receive a second, contralateral implant and to be included in the study, the following criteria were handled; patients with bilateral severe-to-profound hearing loss (≥85dB at 2 and 4 kHz) aged between 5 and 18 years, previously implanted on one side. Patients with cochleovestibular malformations or with signs of intracochlear obliteration on CT-scan that might prevent full insertion of electrode array were excluded. Patients with a high suspicion of having an aplasia of the auditory nerve diagnosed by previous Magnetic Resonance Imaging and with no detectable hearing thresholds were excluded. Otherwise, no limitations were made for any type of etiology of hearing loss. Cases with limited expectations of a second cochlear implant in terms of speech perception results were not qualified for conditional reimbursement of the contralateral implant and therefore not included in this study. These expectations were not based on the distinction of pre- or post-lingual deafness or factors of comorbidity (i.e. developmental delay), but on factors such as limited benefit of the first cochlear implant, sign language as preferred communication mode at home, and minimal abilities to develop speech and language skills.

A full diagnostic evaluation was performed by a multidisciplinary cochlear implantation team (CI-team) embedded within the participating University Medical Centres to assess

eligibility. These multidisciplinary CI-teams consisted of an otorhinolaryngologist, audiologist, speech and language pathologist and a psychologist. Every case underwent an additional, independent review from a second CI-team of another University Medical Centre (based on the patients' file) to make a well-advised decision before giving consent for a second cochlear implant. Parents of patients were counselled about the possible risks of surgery including bilateral vestibular areflexia. The surgical procedure for cochlear implantation was performed according to the standard of care of each cochlear implant team. Cochlear implants from the same manufacturer as the first cochlear implant (Cochlear, Med-El or Advanced Bionics (AB)) were implanted in each individual at the contralateral side.

The study was performed in accordance with the Declaration of Helsinki. Exemption for a full review from the Local Research Ethics Committee (UMCU, Utrecht, The Netherlands) was approved considering the prospective design with the use of pseudo-anonymized data extracted from the regular performance evaluations (local ethics committee No 15–336). Written form of consent was obtained before evaluation. The outcome measurements were included in a coded way in the database during the evaluation moments before and after the second cochlear implant of the individual patients.

### Demographic data collection

Demographic data were collected prior to the implantation of the second cochlear implant (CI2). General characteristics comprised of sex, age at onset of deafness in months, duration of deafness before the first cochlear implantation in months, age of the first and second cochlear implantation in years, level of residual hearing (hearing threshold, in the ear to be implanted with second CI at 250 Hz and 500 Hz) and the pre-implantation use of a hearing aid in the ear to be implanted with CI2 (contralateral of the first cochlear implant (CI1)). Medical characteristics about etiology of hearing loss and comorbidity were collected. Etiology was divided in subgroups of congenital hearing loss with unknown cause (including genetic non-syndromic hearing loss), meningitis, intrauterine infection (including congenital Cytomegalovirus [CMV] infection), inner ear malformations (including Enlarged Vestibular Aqueduct [EVA]), syndromic hearing loss and auditory neuropathy. Syndromic hearing loss includes mutations in DFNA/DFNB or DFNX genes. Comorbidity was categorized in somatic comorbidity, a subgroup of psychiatric and behavioral problems and a subgroup of developmental delay including mild cognitive impairment. Implantation related data were registered; implantation related complications, status of electrode insertion in the cochlea, and the brand of CI device implanted (Cochlear, Med-El or Advanced Bionics (AB)). Education characteristics reported included: type of secondary education of the patient, divided into preparatory vocational or senior general / university preparatory education; type of primary education, classified in special or mainstream education; use of speech therapy; number of spoken languages by patient; maternal education level, divided in secondary (vocational) education level or university (of applied sciences); domestic communication, divided into spoken language either with or without sign language. Frequency of use of the second CI was recorded in the subdivision of daily use, frequent use or non-use, as self-reported by the CI users or parents.

### Outcome assessment

Data of 85 Dutch-speaking children were collected on primary outcomes of speech recognition in quiet and noise, and secondary outcomes of subjective performance of hearing abilities in daily life and speech and language development scores. In all participants these measurements (including residual hearing level assessment) were taken within three months before the second cochlear implantation (baseline measurement) and at 12 months after the second cochlear

implant (postoperative measurement). Comparisons were made between the bilateral CI condition 12 months after implantation compared to 'best aided condition' in unilateral CI1 condition meaning with or without an additional contralateral hearing aid, as chosen by the patient as best condition.

## Outcome measurement

Speech recognition was measured by a speech and language pathologist with the stimulus-repetition task, designed by Bosman and Smoorenburg [23]. During this task the patient had to repeat recorded Dutch monosyllabic words (consonant-vowel-consonant [CVC]), provided by the computer without visual support. These open-set meaningful words were presented in a quiet room at a level of 65dB SPL, first in quiet and subsequently in the presence of steady state noise (with a Signal-to-Noise Ratio [SNR] of 0dB presented at S0˚N0˚). The measurement of speech recognition was performed with two randomly selected CVC word lists, taken from the available pediatric (patients < 12 years of age) or adult (patients ≥ 12 years of age) word list of the NVA (Dutch Audiology Society). Each assessment consists of a list of 11 recorded words (accordingly 33 phonemes), obtained by three conditions: with the first CI, with the second CI and with both CI's. For each condition, the mean percentage correct phoneme score was calculated over the two lists of words. In case of unilateral implantation (before implantation of the second CI) the speech recognition measurements were performed in the 'best aided condition' which means either with only the unilateral cochlear implant or with an additional contralateral hearing aid, as chosen by the patient as best condition. The speech recognition of this unilateral condition was recorded separately in quiet and noise and compared to the speech recognition in quiet or noise measured in the post-implant bilateral condition.

To quantify the receptive vocabulary, the Dutch version of the Peabody Picture Vocabulary Test (PPVT-III-NL) was used, applicable from an age of 2 years and 3 months. The PPVT-III-NL consists of 204 test parts with each 4 pictures, as a multiple-choice test, carried out by a speech and language therapist. The patients had to match the correct picture for each offered spoken word. Subsequently raw scores were recalculated as a standard score, with a population mean of 100 and an +/-1SD of 15, corrected for chronological age in the standard score. The PPVT-III-NL is validated with intelligence tests and other vocabulary tests [24].

The subset "recalling sentences" of the Dutch version of the Clinical Evaluation of Language Fundamentals (CELF-4-NL) [25] was used. The CELF-4-NL is designed to represent the language proficiency in young children of 5 to 18 years old. In this subtest the patient had to listen to spoken phrases uttered by a speech and language therapist, concerning sentences of sequential increasing length and complexity. Thereafter, the patient had to repeat the phrases in the same order and in the correct sentence structure. Outcomes were reported in norm scores, calculated with a reference score, according to age [26]. This subtest of the CELF-4-NL requires higher order complex linguistic skills. The CELF-4-NL is validated with other vocabulary tests, including the PPVT-III-NL and secondly corrected for learning effect [32]. Both tests (PPVT-III-NL and CELF-4-NL) were performed in the preferred or best aided condition as explained above in the unilateral cochlear implant condition.

Information about subjective perceived speech perception, localization abilities and quality of sound was collected with the parent-reported Speech, Spatial and Qualities of Hearing Scale (SSQ) [12]. This questionnaire consists of 30 questions distributed in 3 domains: speech, spatial and qualities of hearing. The questionnaire is adjusted for children by Galvin et al [27] and was translated in Dutch. A scale ranging from 0 (not at all) to 10 (perfectly) is applied to answer the questions about the several aspects of hearing performance of the child. Total score of each domain is divided by 10 to provide comparable outcomes. The questions were mainly

answered by parents of the implanted children or in exceptional cases by the children themselves when they were old enough to rate their own hearing experience.

## Non-user characteristics

In case of perceived non-use of the second cochlear implant one year after surgery, the 12 months outcome measurements of speech recognition with the second cochlear implant and language skills tests were not executed by the inexperience with the second implant. This led to data not at random missing. Though, the patient characteristics (i.e. etiology of deafness) were collected to study this population in more detail and assess the reasons for non-use. The CVC scores and scores in language skills of the unilateral pre-surgery situation were also registered.

## Data analysis

The data of the five participating CI-centers were collected and merged into a uniform database. Medians and quartile percentiles (25th and 75th) or means and standard deviations of the data of users and non-users were reported, depending on normality of the variable. Differences between performance in the unilateral CI and bilateral CI conditions were tested using Wilcoxon signed rank test (for related samples) in case of not normally distributed outcomes, Student's t-tests in case of normally distributed outcomes. Univariable linear regression analyses were performed of the users of the second CI to determine the effect of individual variables on performance outcomes. No multivariable regression will be performed by the expected limited sample size. In each analysis, the level of statistical significance was set at a $p$-value of $< .05$. Details for non-users will be descriptively described because of the low number of non-users. Statistical analyses were performed using SPSS 21.0 for Windows and R [28].

## Results

### Study population

In Fig 1 the number of available data, categorized by outcome, is shown of the 85 patients receiving their second, contralateral cochlear implant between January 2014 and December 2016. This figure displayed the available data excluded the number of missing data. Data did not comply with the assumptions for missing data at random, therefore multiple imputation was not possible. In the study population, 75 (88%) had used their second implant at least 12 months following its implantation. The remainder of the children were non-users of the second device at 12 months post implantation evaluation.

Implantations were performed in five different University Medical Centers: 32 (40%) implantations in Radboud University Nijmegen Medical Center, 22 (28%) implantations in the University Medical Center of Utrecht (UMCU), 14 (16%) in Leiden University Medical Center (LUMC), 11 (13%) in Erasmus Medical Center (EMC) and 6 (7%) in Maastricht University Medical Center (MUMC). In Table 1, the subject characteristics are described.

Table 1 shows the baseline characteristics of the study population. Median age in months at onset of deafness was 0 [range 0–30 months] based on n = 65 (76%) patients (n = 10 with no strict definition of onset of deafness regarded as missing data). Median age at time of the first cochlear implantation was 3 years [range 0–13 years] compared to 12 years [range 5–18 years] at time of the second CI implantation. 35 out of 85 patients (41%) made use of an additional hearing aid at the contralateral side of the first CI before receiving their second implant and therefore regarded as patients with bimodal stimulation. CI2 was implanted in the left ear in 62% (n = 53). Congenital hearing loss with unknown cause, including genetic non-syndromic hearing loss was the main cause (62%, n = 44) of bilateral deafness in the total cohort of

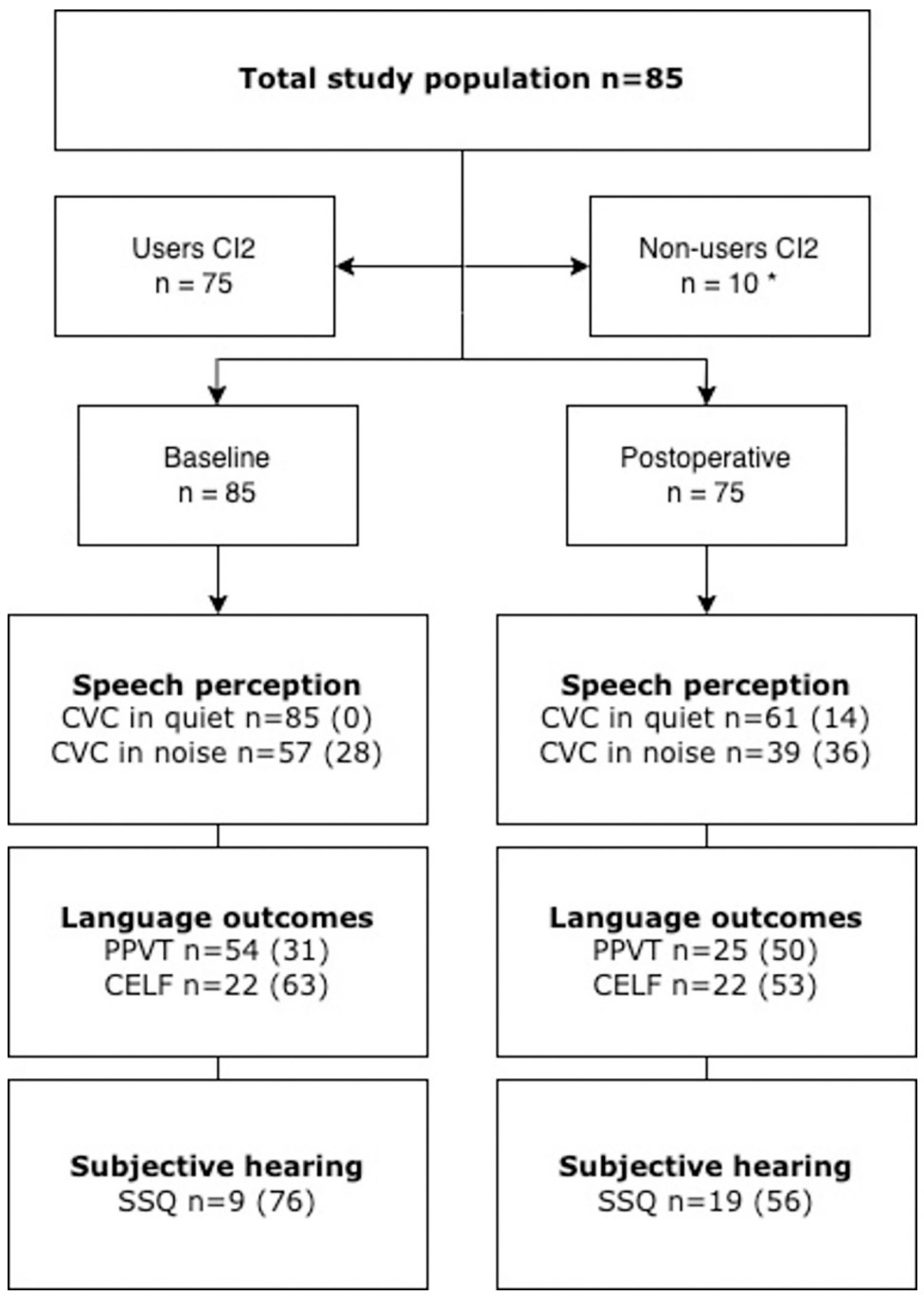

**Fig 1. Flow chart of available test scores.** *Flow chart of study population and numbers of participants in analysis per outcome measure (in brackets missings per outcome). Note*: CVC = consonant-vowel-consonant; SSQ = Speech, Spatial and Qualities of Hearing Scale; PPVT = Peabody Picture Vocabulary Test; CELF = Clinical Evaluation of Language Fundamentals. *For the non-users no postoperative performance outcomes were measured by the non-use of CI2 in daily life*.

implanted children. In two patients an inner ear malformation was present and considered as the etiology of the hearing loss, consisting of an enlarged vestibular aqueduct without malformation of the cochlear duct or modiolus. Implantation related complications occurred in 8 out

**Table 1. Baseline characteristics.**

| Demographic data | User *n (%)*[1] | Non-user *n (%)*[1] |
|---|---|---|
| Number of patients | 75 | 10 |
| **General characteristics** | | |
| Sex | | |
| Male | 34 (45%) | 2 (20%) |
| **Medical characteristics** | | |
| Etiology | | |
| Congenital with unknown cause[2] | 41 (55%) | 3 (30%) |
| Syndromic hearing loss | 12 (16%) | 6 (60%) |
| Meningitis | 3 (4%) | 1 (10%) |
| Intrauterine infection | 2 (3%) | - |
| Inner ear malformation | 2 (3%) | - |
| Auditory neuropathy | 1 (1%) | - |
| Unknown | 14 (19%) | - |
| Comorbidity | | |
| Somatic comorbidity | 10 (13%) | 3 (30%) |
| Psychiatric and behavioural problems | 7 (9%) | - |
| Developmental delay | 2 (3%) | - |
| Age at onset of deafness (median in months)[3] | 0 (0–30) | 0 (0–6) |
| Time of preimplantation deafness CI1 (median in months) | 46 (5–165) | 35 (19–68) |
| **Implantation characteristics** | | |
| Age of CI1 implantation (median in years) | 3 (0–13) | 2 (1–9) |
| Age of CI2 implantation (median in years) | 12 (5–18) | 15 (9–17) |
| Inter-implant interval (median in years) | 8 (1–16) | 12 (11–15) |
| Residual hearing 250 Hz (median in dB) | 85 (35–110) | 85 (55–95) |
| Residual hearing 500 Hz (median in dB) | 95 (55–120) | 95 (65–105) |
| Hearing aid use pre-CI2 | 35 (41%) | 1 (10%) |
| Frequency of use CI2 | | |
| Daily use | 55 (73%) | |
| Frequent use | 20 (27%) | |
| Brand of CI device | | |
| Cochlear | 58 (77%) | 10 (100%) |
| AB | 15 (20%) | - |
| Med-EL | 2 (3%) | - |
| **Education characteristics** | | |
| Type of primary education | | |
| Mainstream education | 30 (40%) | 6 (60%) |
| Special education | 16 (21%) | 2 (20%) |
| Unknown | 29 (39%) | 2 (20%) |
| Type of secondary education | | |
| Preparatory vocational | 15 (20%) | 3 (30%) |
| Senior general / University preparatory | 8 (11%) | 1 (10%) |
| Unknown | 52 (69%) | 6 (60%) |
| Speech therapy | 11 (15%) | 2 (20%) |
| Unknown | 64 (85%) | - |
| Spoken languages by patient | 52 (69%) | 10 (100%) |
| Native language | 9 (12%) | - |
| Multiple languages | 14 (19%) | - |
| Unknown | | |
| Domestic communication | 44 (59%) | 6 (60%) |
| Spoken language without sign language | 13 (17%) | 4 (40%) |
| Spoken language with sign language | 3 (4%) | - |
| Single sign language | 15 (20%) | - |
| Unknown | | |

(*Continued*)

**Table 1.** (Continued)

| Demographic data | User *n (%)*[1] | Non-user *n (%)*[1] |
|---|---|---|
| Maternal education level | 26 (35%) | 6 (60%) |
| Secondary (vocational) education | 11 (15%) | 4 (40%) |
| University (of applied sciences) | 38 (51%) | - |
| Unknown | | |

*Note*: CI2 = second cochlear implant.

1 Reported in median (range) when noted.

2 Including genetic non-syndromic hearing loss.

3 Based on the available data (missing excluded data).

of 85 cases consisting of 6 patients (8.5%). This required revision surgery to drill-out the implant receiver-site because of receiver migration (n = 4) or an incomplete insertion of the electrode array (n = 2). Remaining complications included fever the day after surgery without the need of antibiotics (n = 1), and a defect of the external ear canal that occurred during drilling of the mastoid which was repaired during the same surgical procedure with a bonechip (n = 1). Fig 1 shows the numbers of available data for the different outcomes.

## Speech recognition and speech and language outcomes

Table 2 contains the results of speech recognition outcomes in quiet and noise and language outcomes before (baseline) and 12 months after second cochlear implantation (postoperative measurement). With bilateral CI, the median phoneme score in quiet at the 12 months evaluation showed a significantly higher score compared to the unilateral implanted situation for the total group of recipients (postoperative CI2 bilateral condition 96% [90–98] vs baseline best aided CI1 condition 91% [85–96]; $p = 0.02$) and for the subgroup of recipients with unilateral stimulation before CI2 (postoperative CI2 bilateral CI 96% [89–98] vs unilateral stimulated baseline CI1 condition 90% [84–96]; $p < 0.01$). Also, the phoneme score in noise [SNR 0dB] 12 months after the second implantation showed a significant higher result in comparison to the best aided unilateral implanted situation for the total group as well for the group of bimodal stimulation recipients (respectively median score of 66% [57–75] vs 54% [47–71]; $p = 0.01$ and median score of 66% [58–75] vs 51% [47–74]; $p = 0.03$). Receptive vocabulary and recalling sentences, showed no statistical differences when comparing unilateral to bilateral implantation.

## SSQ perceived hearing abilities in daily life

Table 3 presents the outcomes for the three separate domains of the SSQ (speech, spatial, qualities). The speech and spatial domains and the total SSQ score (total mean score of the 3 domains) did not significantly improve comparing the results of the daily situation of the unilateral implanted condition versus the post-operative measurement 12 months post second cochlear implantation. The sound quality domain showed a statistically significant higher rating for the bilateral CI situation (6.6 [5.0–7.1]) in comparison to the unilateral CI scores (5.0 [3.4–6.7]; $p = 0.041$).

## Non-user characteristics

Ten out of 85 (12%) second cochlear implant recipients were registered as non-user at the 12 months evaluation assessment (Table 1). These patients either did not experience added value of the second device (n = 4, 40%), were not motivated for revalidation for unknown reasons

**Table 2. Results of speech recognition and speech and language development outcome.**

| Condition Outcome | Baseline CI1 condition | | Baseline CI2 unilateral condition [4] | | Postoperative CI2 unilateral condition | | Postoperative CI2 bilateral condition | | Z-statistic p-value [5] |
|---|---|---|---|---|---|---|---|---|---|
| | n | Median (IQR) | n | Median (IQR) | n | Median (IQR) | n | Median (IQR) | |
| Total | **85** | | **85** | | **75** | | **75** | | |
| CVC score in quiet (daily life) [1] *Missing* | 85 *0* | 91 (85–96) | 30 *55* | 45 (20–69) | 36 *39* | 65 (43–80) | 61 *14* | 96 (90–98) | Z = 3.01 **0.02** |
| Bimodal | 35 | 96 (86–98) | 23 | 52 (21–68) | 13 | 77 (65–83) | 29 | 96 (90–99) | Z = 1.10 0.27 |
| Unilateral | 44 | 90 (84–96) | NA | NA | 17 | 44 (16–63) | 28 | 96 (89–98) | Z = 2.70 <**0.01** |
| Missing [2] | 6 | | 6 | | 6 | | 4 | | |
| CVC score in noise (daily life) [3] *Missing* | 57 *28* | 54 (47–71) | | | 9 *66* | 36 (28–50) | 39 *36* | 66 (57–75) | Z = 2.80 **0.01** |
| Bimodal | 29 | 51 (47–74) | | | 6 | 38 (26–47) | 22 | 66 (58–75) | Z = 2.18 **0.03** |
| Unilateral | 28 | 55 (46–68) | | | 3 | 30(*) | 17 | 63 (54–76) | Z = 1.37 0.17 |
| Missing [2] | 0 | | | | 0 | | 0 | | |
| PPVT standardscore *Missing* | 54 *31* | 89 (79–99) | | | | | 25 *50* | 92 (79–105) | Z = 1.61 0.16 |
| Bimodal | 27 | 96 (85–104) | | | | | 12 | 96 (88–104) | Z = 1.07 0.29 |
| Unilateral | 27 | 80 (63–96) | | | | | 13 | 85 (65–109) | Z = 1.12 0.26 |
| Missing [2] | 0 | | | | | | 0 | | |
| CELF normscore *Missing* | 22 *63* | 4.5 (1.0–8.0) | | | | | 22 *53* | 6.0 (2.0–8.0) | Z = 1.14 0.32 |
| Bimodal | 7 | 8.0 (5.0–9.0) | | | | | 10 | 7.0 (4.0–9.0) | Z = 0.65 0.52 |
| Unilateral | 15 | 3.0 (1.0–5.0) | | | | | 12 | 4.0 (1.0–7.0) | Z = 0.96 0.34 |
| Missing [2] | 0 | | | | | | 0 | | |

Results of speech recognition and speech and language development outcome of the baseline situation with first CI (CI1) without (unilateral stimulation) or with contralateral hearing aid (bimodal stimulation)('baseline CI1 condition'), the baseline perception of the ear of second CI ('baseline CI2 unilateral condition'), the post-operative perception of the ear with second CI ('Postoperative CI2 unilateral condition'), and the post-operative perception measured with both implants ('Postoperative CI2 bilateral condition'). *Outcomes were scored for all participants at baseline (n = 85). Postoperative scores depict the outcomes of participants registered as users of CI2 (n = 75).*

*Note*: CI = Cochlear Implant; Scores displayed by the medians and 25th and 75th percentiles (IQR, in parentheses).

* IQR not applicable by low numbers of participants. The exact number of data used for analysis is specified with *n*.

1 In 'best aided' hearing situation: With (bimodal stimulation) or without (unilateral stimulation) a contralateral hearing aid to the CI, as used in daily life situation by the patient.

2 Missing data of use of hearing aid pre-operative in the ear of the second CI.

3 Measured with [SNR 0dB].

4 Speech perception in quiet, of the bimodal stimulated children, was measured at baseline with hearing aid in the ear of the second CI.

5 p-value of the Wilcoxon signed-ranks test, calculated by comparison of outcomes of participants 'postoperative CI2 bilateral condition' and 'Baseline CI1 condition'.

(n = 3, 30%), suffered from incoherent hearing performance with both CI's during simultaneously use (n = 2, 20%) or suffered from pain complaints wearing the second CI (n = 1, 10%). 6 patients (60%) of this non-user group had etiology of syndromic hearing loss (i.e. Pendred

Table 3. Results of Speech, Spatial and Qualities of Hearing Scale (SSQ).

| Outcome | Baseline n = 9 | | Post-operative n = 19 | | Z-statistic p-value |
|---|---|---|---|---|---|
| | Median | | Median | | |
| Speech domain | 5.5 | (3.7–5.8) | 6.0 | (5.4–6.3) | Z = 1.19 0.24 |
| Spatial domain | 3.4 | (0.7–6.7) | 4.9 | (3.0–6.4) | Z = -1.01 0.31 |
| Qualities domain | 5.0 | (3.4–6.7) | 6.6 | (5.0–7.1) | Z = 2.04 0.04 |
| Total SSQ score | 4.9 | (2.7–6.4) | 5.9 | (5.0–6.6) | Z = 1.52 0.13 |

*Note*: Scores displayed by the medians and 25th and 75th percentiles (in parentheses).

syndrome), 1 patient (10%) had meningitis and 3 patients (30%) had congenital hearing loss with unknown cause. Secondly, 3 patients (30%) had somatic comorbidity (Usher syndrome (2), vitamin deficiency (1)). In two (20%) patients (one with ossification (meningitis) and one with cochlear anomaly (Pendred syndrome)) full insertion of the electrode array was not achieved. After revision surgery (full insertion achieved) there was no increase in (bilateral) subjective speech perception compared to the situation with CI1 only, which resulted in non-use of CI2 by all 3 patients. The level of residual hearing before implantation of CI2 was similar with the user group. The median age of the first cochlear implantation in the non-user group was 2 years and of the CI2 implant 15 years (user group respectively 3 years and 12 years). The differences between the non-users group and the group of users were: a median inter-implant interval of 147 months (range 133–178 months) versus 96 months (range 13–192 months), median best-aided preoperative CVC score in quiet of 90% (range 36–100) versus 91% (range 85–96), median best-aided preoperative CVC score in noise of 48% (range 18–76) versus 54% (range 47–71), median preoperative PPVT WBQ score of 85 (range 55–96) versus 89 (range 79–99) and a median preoperative CELF norm score of 4.0 (range 1.0–11.0) versus 4.5 (1.0–8.0) respectively. The 12 months evaluation scores were not available, since this group did not use the second cochlear device.

## Linear regression analysis

The results of the univariable linear regression analysis are presented in Table 4. Frequent use was significantly negatively associated with the postoperative CVC outcome in noise ($\beta$ = -12.05; 95% CI -22.17- -1.94; $p$ = 0.02) compared to daily use of the second CI. Special education as type of primary school was significantly associated with lower PPVT (age corrected) scores 12 months after CI2 ($\beta$ = -21.30, 95% CI -37.36- -5.25; $p$ = 0.01). A higher secondary education level ($\beta$ = 4.30; $p$ = 0.02) was significantly positively associated with the CELF outcome in the bilateral CI condition. Residual hearing level in the ear to be implanted with the second implant, and age at the time of the second implantation were not significantly associated with the performance outcomes (Table 4). Similar results were seen for the inter-implant interval which was not significantly related to the performance outcomes after CI2 for the studied cohort in total, or for the individual groups of unimodal and bimodal stimulated recipients. In the Supporting information the results are visualized for the CVC outcomes in relation to the inter-implant interval for the bimodal and unilateral users separately.

**Table 4. Univariable linear regression analysis of outcomes of CI2 users 12 months after the second cochlear implant (CI2).**

| Variable | CVC in quiet | | | | CVC in noise | | | | PPVT[1] | | | | CELF[1] | | | |
|---|---|---|---|---|---|---|---|---|---|---|---|---|---|---|---|---|
| | n | Mean (SD) | β (95% CI) | p | n | Mean (SD) | β (95% CI) | p | n | Mean (SD) | β (95% CI) | p[2] | n | Mean (SD) | β (95% CI) | p[2] |
| Sex Male (ref) / *Missing* Female / *Missing* | 268 / 356 | 94.4 (6.0) / 91.3 (11.9) | -3.12 (-8.22–2.00) | 0.23 | 17 / 17 / 22 / 19 | 61.2 (22.3) / 65.8 (9.6) | 4.64 (-6.04–15.32) | 0.38 | 13 / 21 / 12 / 29 | 95.2 (15.5) / 89.3 (25.0) | -5.98 (-23.06–11.10) | 0.48 | 13 / 219 / 32 | 7.0 (3.7) / 3.6 (2.4) | -3.44 (-6.39--0.50) | **0.02** |
| Age at CI2 (years) / *Missing* | 61 / 14 | 92.6 (9.9) | -0.39 (-1.01–0.23) | 0.21 | 39 / 36 | 64.2 (16.3) | -0.98 (-2.30–0.34) | 0.14 | 25 / 50 | 93.5 (20.0) | -0.16 (-2.77–2.46) | 0.90 | 22 / 53 | 5.8 (3.6) | 0.43 (-0.03–0.90) | 0.06 |
| Residual hearing level[3] / *Missing* | 53 / 22 | 92.6 (10.5) | 0.11 (-0.09–0.31) | 0.28 | 30 / 45 | 64.0 (15.7) | -0.03 (-0.51–0.44) | 0.89 | 17 / 58 | 96.9 (19.2) | -0.34 (-1.46–0.79) | 0.54 | 14 / 61 | 6.6 (3.6) | 0.12 (-0.13–0.37) | 0.32 |
| Inter-implant interval (years) / Unilateral / *Missing* / Bimodal / *Missing* | 53 / 28 / 7 / 29 / 5 | 92.6 (10.5) / 93.9 (4.8) / 93.6 (6.9) | 0.40 (-0.25–1.06) / -0.23 (-0.75–0.28) / 0.22 (-0.79–1.22) | 0.22 / 0.36 / 0.66 | 39 / 17 / 18 / 22 / 12 | 63.8 (16.3) / 64.9 (13.3) / 63.0 (18.5) | 0.98 (-0.68–2.63) / 0.62 (-1.45–2.68) / 1.49 (-1.68–4.65) | 0.24 / 0.53 / 0.34 | 25 / 13 / 22 / 12 / 22 | 92.4 (20.4) / 88.2 (26.2) / 96.8 (11.0) | -1.37 (-3.86–1.12) / -1.29 (-5.87–3.29) / -0.48 (-3.42–2.47) | 0.66 / 0.55 / 0.73 | 22 / 11 / 24 / 11 / 23 | 5.6 (3.6) / 4.2 (3.1) / 7.0 (3.6) | -0.13 (-0.64–0.39) / -0.07 (-0.74–0.60) / 0.19 (-0.81–1.18) | 0.61 / 0.82 / 0.68 |
| Hearing aid before CI2 / Bimodal / *Missing* / Unilateral (ref) / *Missing* | 29 / 5 / 28 / 7 | 93.5 (6.8) / 93.8 (4.7) | -0.30 (-3.42–2.81) | 0.85 | 22 / 12 / 17 / 18 | 63.0 (18.5) / 64.9 (13.3) | -1.93 (-12.70–8.85) | 0.72 | 12 / 22 / 13 / 22 | 96.3 (11.0) / 88.2 (26.2) | 8.60 (-8.27–25.47) | 0.30 | 11 / 23 / 11 / 24 | 7.0 (3.7) / 4.2 (3.1) | 2.82 (-0.21–5.85) | 0.07 |
| Secondary education level / Sr. general / Univ. prep. / *Missing* / Prep. Vocational (ref) / *Missing* | 7 / 1 / 150 | 93.0 (6.1) / 91.9 (7.3) | 1.13 (-5.54–7.81) | 0.73 | 6 / 2 / 11 / 4 | 58.7 (22.3) / 64.5 (23.1) | -5.79 (-30.52–18.94) | 0.63 | 5 / 3 / 8 / 7 | 107.8 (12.7) / 96.5 (8.5) | 11.30 (-1.53–24.13) | 0.08 | 5 / 3 / 8 / 7 | 9.8 (3.1) / 5.5 (2.3) | 4.30 (1.03–7.57) | **0.02** |
| Type of primary school / Special / *Missing* / Mainstream (ref) / *Missing* | 124 / 49 / 10 | 93.3 (4.6) / 92.5 (10.8) | 0.76 (-5.67–7.19) | 0.81 | 11 / 5 / 28 / 31 | 64.7 (13.2) / 63.4 (17.5) | 1.30 (-10.59–13.19) | 0.83 | 8 / 8 / 17 / 42 | 77.9 (15.1) / 99.2 (19.3) | -21.30 (-37.36--5.25) | **0.01** | 7 / 9 / 15 / 44 | 4.3 (2.8) / 6.2 (3.9) | -1.91 (-5.34–1.51) | 0.26 |
| Frequency of CI2 use / Daily use (ref) / *Missing* / Frequent use / *Missing* | 430 / 182 | 92.7 (11.1) / 92.6 (6.6) | -0.04 (-5.65–5.57) | 0.99 | 23 / 31 / 16 / 4 | 68.7 (11.9) / 56.7 (19.2) | -12.05 (-22.17--1.94) | **0.02** | 13 / 41 / 12 / 8 | 91.3 (25.0) / 93.5 (15.1) | 2.20 (-15.05–19.44) | 0.80 | 10 / 44 / 12 / 8 | 4.1 (3.1) / 6.8 (3.6) | 2.73 (-0.33–5.79) | 0.08 |

(*Continued*)

**Table 4.** (Continued)

| Variable | CVC in quiet | | | | CVC in noise | | | | PPVT¹ | | | | CELF¹ | | | |
|---|---|---|---|---|---|---|---|---|---|---|---|---|---|---|---|---|
| | n | Mean (SD) | β (95% CI) | p | n | Mean (SD) | β (95% CI) | p | n | Mean (SD) | β (95% CI) | p ² | n | Mean (SD) | β (95% CI) | p ² |
| **Maternal education level** | | | | | | | | | | | | | | | | |
| Secondary voc. (ref) | 20 | 89.9 (14.9) | 6.60 (-3.36–16.56) | 0.19 | 13 | 67.6 (16.0) | -1.32 (-13.00–10.37) | 0.82 | 10 | 93.6 (28.2) | -8.60 (-55.25–38.05) | 0.69 | 7 | 4.9 (3.4) | 0.64 (-5.58–6.86) | 0.81 |
| *Missing* | *6* | | | | *13* | | | | *16* | | | | *19* | | | |
| University | 10 | 96.5 (4.5) | | | 10 | 66.3 (8.7) | | | 2 | 85.0 (12.7) | | | 2 | 5.5 (2.1) | | |
| *Missing* | *3* | | | | *3* | | | | *11* | | | | *11* | | | |
| **Speech therapy** | | | | | | | | | | | | | | | | |
| Yes | 7 | 81.7 (23.9) | -11.85 (-24.51–0.81) | 0.07 | 5 | 46.4 (28.1) | -17.43 (-41.73–6.86) | 0.15 | 5 | 77.4 (12.4) | -17.31 (-37.27–2.64) | 0.09 | 5 | 3.4 (2.3) | -3.37 (-7.19–0.45) | 0.08 |
| *Missing* | *4* | | | | *6* | | | | *6* | | | | *6* | | | |
| No (ref) | 16 | 93.6 (4.9) | | | 12 | 63.8 (18.4) | | | 14 | 94.7 (19.6) | | | 13 | 6.8 (3.7) | | |
| *Missing* | *0* | | | | *4* | | | | *2* | | | | *3* | | | |
| **Spoken languages** | | | | | | | | | | | | | | | | |
| Multiple | 7 | 93.6 (4.5) | 1.29 (-7.54–10.12) | 0.77 | 7 | 62.9 (18.1) | -1.14 (-15.08–12.80) | 0.87 | 4 | 97.3 (25.7) | 5.82 (-17.58–29.23) | 0.61 | 4 | 7.8 (5.4) | 2.64 (-1.46–6.74) | 0.19 |
| *Missing* | *2* | | | | *2* | | | | *5* | | | | *5* | | | |
| Single (ref) | 43 | 92.3 (11.4) | | | 32 | 64.0 (16.2) | | | 21 | 91.4 (19.9) | | | 18 | 5.1 (3.1) | | |
| *Missing* | *9* | | | | *20* | | | | *31* | | | | *34* | | | |

*Note*: (BI)CI = (Bilateral) Cochlear Implant; *n* = number of available biographic data; β = standardized regression coefficients; SD = Standard Deviation; *p* = p-value.

1 Reference (norm) score, according to age.

2 Results marked in bold showed a statistically significant association ($p < 0.05$).

3 Pre-implantation scores of the ear to be implanted with CI2 (mean threshold of 250-500Hz).

## Discussion

### Key findings and comparison with other studies

In this multicenter prospective trial, we evaluated the benefit (at 12 months post-surgery) of a second cochlear implant in children at the age of five to eighteen years with severe to profound hearing loss after previous unilateral implantation. We demonstrated a significant and relevant improvement in speech perception in both quiet and noise scores at 12 months post-implantation compared with the (best aided) unilateral CI situation. Generally, speech recognition scores in quiet were considered to be good, even with a unilateral cochlear implant, limiting the maximum improvement in scores especially for those with bimodal sequential stimulation as in accordance with previous literature [29,30]. Though in this study, a significant increase of speech recognition in noise after the second CI was demonstrated, combining the unilateral and bilateral sequential stimulation recipients, even with a mean inter-implant interval of 8 years and a median age of 12 years [*SD* = 4,0 years] at second implantation. This outcome adds value to existing literature describing benefits in situations with shorter intervals and younger study populations [3,16,21,31].

In our study, no statistical significant difference was seen in receptive vocabulary and sentence recalling between the bilateral CI situation and unilateral CI condition. The median bilateral PPVT and CELF scores yielded a large standard deviation possibly induced by a heterogenic study population or a large variety in verbal intelligence [32,33], which was not tested in our study. Some children performed poorly and therefore received speech therapy to improve and stimulate their language abilities, some children did not need this support and performed better on outcomes, as shown in Table 4. In addition, it has been reported by Hay-McCutcheon et al. [34] that the age of implantation does not have a significant impact on the

receptive and expressive language abilities of children aged nine years and older. For this reason, the effects on language performance in our study population aging 5–18 years is expected to be limited, as most of the children were older than 9 years old at time of their second cochlear implantation. In addition, the effect of the bilateral cochlear situation in our study was evaluated at 12 months after implantation of the second cochlear implant. Sparreboom et al. [19] found a significant bilateral benefit on receptive vocabulary after 5 years of bilateral implant experience in older children. One could therefore hypothesize that the duration of bilateral implant use of 12 months in the current study group might be too short to expect a benefit in speech and language development. Besides that, the PPVT scores corrected for age in the unilateral CI situation (median of 89 [79–99]) were already adequate before the second cochlear implantation. As the scores for the subtest recalling sentences was on average below that of their peers with normal hearing, one would therefore expect that the improvement in language skills might be eventually seen in de the more complex linguistic skills.

Overall, no significant benefit was measured on the subjective outcome measured by the SSQ. When analyzing the domains separately a significant increase on the domain of 'qualities' in the bilateral CI condition compared to the unilateral CI condition was found. This outcome repletes the improvement of hearing quality with the second cochlear implant in comparison to the unilateral implanted condition.

In our study, no effect was seen by the age of implantation of the second CI or the length of the inter-implant interval on speech perception performance or development by univariable linear regression analysis, even when analyzing unilateral and bimodal stimulated users separately. This could be explained by the eligibility criteria and characteristics of the studied cohort, combining cases with and without progressive hearing loss. Though, longer time of deafness before the first cochlear implant as well as a higher age at time of the second implantation have previously demonstrated to be negatively correlated with these outcomes [31,35,36]. This could be contributed to a specific sensitive period for auditory development on the deprived side [37]. Also, from several studies it seems that limited device use besides a longer inter-implant delay has a detrimental effect on auditory brainstem and subsequent cortical maturation [38–40]. However, in these particular studies, no children with progressive hearing loss were included. As in our study children with progressive hearing loss were included, the negative effect of a longer inter-implant delay might have been less obvious. Recently, Illg et al. demonstrated the impact of the length of the inter-implant interval on the benefit of sequentially implanted bilateral CI. The authors suggested a maximum interval of up to four years for receiving their second implant for children implanted with their first CI under the age of 4 [41]. The older the children were at first implantation, the shorter the inter-implant interval had to be to favor good speech comprehension results. The non-users in our study population showed a greater inter-implant interval in comparison with the user group, which could therefore have a negative effect on outcome. Interesting issue is to discuss the question why the non-users became non-user of the second CI. A possible reason could be the timing of implantation during (early) adolescence which could be related to difficulties with the acceptation of a new (hearing) situation. Additionally, only one out of ten (10%) used a hearing aid prior to CI2 in this non-user group compared to 35 out of 85 (41%) in the user group. As a result, motivation or expectations of outcomes might have been different between these children. Another explanation could be the incoherent hearing performance experienced with both CI's during simultaneously use or pain complaints [42]. Easwar reported that higher speech perception scores were associated with longer everyday CI use and CI experience ($p < 0.05$). Secondly, they described that asymmetry in speech perception between both CIs decreased with consistent everyday use of the second CI ($p < 0.05$) [21]. These results have to be taking in account in the clinical counseling before implantation of a second sequential CI.

## Strengths and limitations

Strength of this study is the unique study population of sequentially implanted children with the first implant at a relatively early age (median 3 years) and their second implant between 5–18 years of age, considered the study population also contains children with progressive hearing loss. By the multicenter design and prospective data collection the outcome of this study will strengthen the evidence of the benefit of sequential cochlear implantation with larger implant-intervals in children. In addition, in this study we were able to combine objective and subjective outcomes of hearing abilities. This makes the results applicable for use in a clinical setting and useful for counseling patients and their parents about the benefit of this intervention, even with children at an older age.

A limitation of this study is the number of missing data, i.e. the SSQ scores, since not every participating CI-center was using this type of evaluation. Based on the studied sample size and anticipated missing data we were not able to perform multivariable regression analysis to calculate outcomes whilst controlling for covariates. Moreover, due to the multicenter data collection, small deviations in test setups of speech recognition outcomes can be expected. Thirdly, in this study, the used eligibility criteria were set based on the criteria advocated by the Dutch health care institute for conditional reimbursement at that time. These criteria have impact on the generalizability of the outcome of the study. Another issue is the 'early' age of the first CI implantation in our study population. The median age at first implantation was 3 years old but with a range until 13 years old. This can be due to variable reasons, i.e. included patients with progressive hearing loss at older age in the study population, comorbidity or domestic situation. Because of the non-user status of CI2 of 10 out of 85 patients, the overall outcome of the second cochlear implant measured in remaining group could be overestimated. Moreover, in the current study we were not able to analyze the influence of the daily device use by lack of datalogging features in the used cochlear implants. Therefore, the relation between daily CI use and performance outcomes could not be analyzed as potential factor of influence. Lastly, we were not able to test localization skills which could be of importance to analyze benefits in more detail. Only the spatial domain of the SSQ was assessed demonstrating no significant bilateral benefit, possibly caused by the limited number of measurements.

## Future recommendation

Because of the lack of objective measures in the postoperative CI2 situation for the non-user group, we were unable to analyze the contributing factors responsible for this result. The non-user group contained children with specific conditions (e.g. Usher syndrome) eligible for a second cochlear implant, notwithstanding a limited expectation of increase of speech perception of the multidisciplinary CI-team, but indicated as eligible by 'benefit of the doubt'. For future analysis it would be interesting to account for expectations of outcome and prognostic social-emotional factors possible influencing the frequency of CI use [13,43,44]. Secondly, including a larger cohort to compare outcomes in unilateral and bimodal stimulated sequential cochlear implants users and including longer period of follow-up after sequential cochlear implantation could provide knowledge about the long-term outcomes in receptive vocabulary and recalling sentences for these children.

## Conclusion

This study demonstrated, statistically significantly and clinical relevant, better speech recognition in quiet and noise, regardless of a greater inter-implant interval in children receiving their first cochlear implant at an relatively early age and then sequentially implanted aged 5 to 18 years. Bilateral cochlear implantation was not associated with positive effects on receptive

vocabulary or sentence recalling after 12 months of BICI use. The subjective sound quality was statistically significant higher rated for the bilateral CI situation in comparison to the unilateral CI scores. Out of the participants 12% was found to become non-user within 1 year after CI2, possibly due to limited residual hearing and a larger inter-implant interval compared to the user-group.

## Supporting information

**S1 Fig. Regression line between speech perception scores (CVC) in quiet in the bilateral CI situation and the inter-implant interval.** β unilateral = -0.23 ($n$ = 29); $p$ = 0.22. β bimodal = 0.08 ($n$ = 28); $p$ = 0.66. *Note*: CI1 = first cochlear implant; CI2 = second cochlear implant CVC = consonant-vowel-consonant (speech perception).
(DOCX)

**S2 Fig. Regression line between speech perception scores (CVC) in noise in the bilateral situation and the inter-implant interval.** β unilateral = 0.44 ($n$ = 17); $p$ = 0.63. β bimodal = 1.49 ($n$ = 21); $p$ = 0.34. *Note*: CI1 = first cochlear implant; CI2 = second cochlear implant CVC = consonant-vowel-consonant (speech perception).
(DOCX)

## Acknowledgments

We would like to thank all participating centers for their contribution. We thank W.J. Smilde for her contributions to the study protocol, and G.G.J Ramakers for the initiation of the data collection.

## Author Contributions

**Conceptualization:** J. H. M. Frijns, M. P. van der Schroeff, M. J. W. Lammers, W. Grolman, A. L. Smit.

**Data curation:** W. J. Kleijbergen, M. Sparreboom, G. de Koning, H. W. Helleman, P. P. B. M. Boermans, J. L. Vroegop, E. E. J. Gelders, E. L. J. George, M. J. W. Lammers.

**Formal analysis:** W. J. Kleijbergen, I. Stegeman.

**Investigation:** A. L. Smit.

**Methodology:** W. J. Kleijbergen, M. Sparreboom, I. Stegeman, A. L. Smit.

**Project administration:** W. J. Kleijbergen, M. J. W. Lammers, A. L. Smit.

**Software:** I. Stegeman.

**Supervision:** M. Sparreboom, E. A. M. Mylanus, P. P. B. M. Boermans, J. H. M. Frijns, M. P. van der Schroeff, E. L. J. George, I. Stegeman, A. L. Smit.

**Validation:** G. de Koning, H. W. Helleman, I. Stegeman, A. L. Smit.

**Visualization:** W. J. Kleijbergen.

**Writing – original draft:** W. J. Kleijbergen, A. L. Smit.

**Writing – review & editing:** W. J. Kleijbergen, M. Sparreboom, E. A. M. Mylanus, H. W. Helleman, I. Stegeman, A. L. Smit.

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
