## [Decision Letter · Decision Letter 0]

24 Feb 2021

PONE-D-20-37958

Benefit of sequential bilateral cochlear implantation in children between 5 to 18 years old: a prospective cohort study

PLOS ONE

Dear Dr. Kleijbergen,

Thank you for submitting your manuscript to PLOS ONE. After careful consideration, we feel that it has merit but does not fully meet PLOS ONE’s publication criteria as it currently stands. Therefore, we invite you to submit a revised version of the manuscript that addresses the points raised during the review process.

We look forward to receiving your revised manuscript.

Kind regards,

Andreas Buechner, PhD

Academic Editor

PLOS ONE

Journal Requirements:

Reviewers' comments:

Reviewer's Responses to Questions

**Comments to the Author**

1. Is the manuscript technically sound, and do the data support the conclusions?

Reviewer #1: Partly

Reviewer #2: Partly

2. Has the statistical analysis been performed appropriately and rigorously? 

Reviewer #1: No

Reviewer #2: No

3. Have the authors made all data underlying the findings in their manuscript fully available?

Reviewer #1: Yes

Reviewer #2: Yes

4. Is the manuscript presented in an intelligible fashion and written in standard English?

Reviewer #1: No

Reviewer #2: Yes

5. Review Comments to the Author

Reviewer #1: Summary:

The manuscript presents a multi-centre prospective study of the benefit of sequential cochlear implantation on speech, language, and quality of life outcomes. The study examines a group (n=85) or children (5 – 18 yrs old) who have experienced a long delay between receiving their first and second implant.

This is a revised version of a previous manuscript. Whilst the authors have made significant changes to the manuscript, there are several major concerns regarding the analysis that need to be addressed before the main claims of the paper can be fully supported.

What are the main claims of the paper and how significant are they for the discipline?

The manuscript reports significant improvements in speech recognition in quiet and in noise following receipt of the second implant. The authors claim no evidence of a relationship between outcomes and key predictors of interest such as inter-implant delay and age at implantation.

In the current form, these findings are not significant for the discipline due to my concerns outlined below. However, the data are important and would be of value to the field if analysed and presented more appropriately.

Are the claims properly placed in the context of the previous literature? Have the authors treated the literature fairly?

Whilst the authors have made improvements to the introduction, I am surprised to see that key work is still not considered here. The authors state that “so far, studies of sequentially implanted children included populations with a mean age between 5 to 10 years” and that outcomes “in older children and adolescents and the influence of preoperative residual hearing is not clear from the literature so far”. This is not the case and the authors are clearly unaware of the available body of literature. Of particular relevance, please see key studies from Dr Karen Gordon’s lab such as Easwar et al., (2018) and Polonenko et al., (2018). These papers include data from older children and adolescents and demonstrate the bilateral benefits of sequential implantation in quiet and noise. Furthermore, they demonstrate the relationship between inter-implant delay and pre-operative residual hearing (amongst other key factors such as age and daily CI use).

It is not until the methods section that is becomes apparent that some children in the sample used a unilateral CI only and so experienced unilateral deprivation/stimulation prior to receiving the second CI. Whereas others (45% stated in Table 1) wore a contralateral hearing aid and so experienced bimodal input (i.e. bilateral access to sound from a combination of electric and acoustic input). These are two very different groups and this should be highlighted to the reader from the very beginning of the manuscript. Literature on bimodal users should also be reviewed. The definition of inter-implant delay therefore also differs for these groups and should be explicated for the non-expert. I.e. duration of unilateral deprivation/stimulation, vs duration of bimodal stimulation.

Do the data and analyses fully support the claims? If not, what other evidence is required?

No. Given that almost half of the sample were bimodal CI and HA users and the other half were unilateral CI users, this should be treated as a factor in the analyses. Bimodal sequential users are likely to receive more benefit from a second implant as their pathways have received acoustic stimulation vs no stimulation in the unilateral CI users. There is evidence that bimodal sequential users gain similar bilateral advantage to that of simultaneous bilateral CI users. Therefore, data for unilateral sequential and bimodal sequential should be investigated to understand if there are group differences here as this could be driving the main result reported here.

Age at time of speech perception testing should be entered as a covariate as younger children can have worse speech reception thresholds, and any developmental effects should be considered when reporting changes in children’s speech and language outcomes over a 12-month period.

I am concerned that the objective of the manuscript is to determine the benefit of a second sequential CI, yet “limited expectations in terms of speech perception results were an exclusion factor to qualify for this study”. Therefore, the inclusion criteria could bias the results by including only those children who were expected to show improvements in speech perception. It is understandable to exclude children based on non-verbal communication/inability to undergo speech perception outcome measures. However, if they received little benefit from the first implant, is that to say that they would not benefit from bilateral input?

Non-use of the second CI is an important outcome as it suggests that the child gained no benefit from sequential implantation. The outcome data from these children is not missing at random – it is missing because they have low benefit. The authors acknowledge this and provide information about this (lines 330-351). However, rather than repeating summary statistics already listed in Table 1, it would be useful if the authors provided information regarding any significant differences between the non-users and the user groups. Of note, the non-users appear older at CI2, have a longer inter-implant interval, and only 10% compared to 45% of users are bimodal hearing aid users prior to CI2. The implications of this on the current findings and main claims should be considered and unpacked further.

The authors state that there are no significant associations between outcomes and certain predictors of interest. However, they fail to identify and discuss some clear trends that are approaching significance. It is likely that some associations have not been detected due to my aforementioned concerns of the mixed group of bimodal sequential and unilateral sequential users, as well as age at time of testing that should be treated as a covariate. Indeed, Table 4 shows that hearing aid use is a predictive factor of language outcomes. The relationships presented in Table 4 should be examined in these two groups separately. Data should be presented visually in the form of scatterplots with regression lines, for example, so that any trends and their importance can be assessed by the reader.

PLOS ONE encourages authors to publish detailed protocols and algorithms as supporting information online. Do any particular methods used in the manuscript warrant such treatment? If a protocol is already provided, for example for a randomized controlled trial, are there any important deviations from it? If so, have the authors explained adequately why the deviations occurred?

No

If the paper is considered unsuitable for publication in its present form, does the study itself show sufficient potential that the authors should be encouraged to resubmit a revised version?

Yes.

Are original data deposited in appropriate repositories and accession/version numbers provided for genes, proteins, mutants, diseases, etc.?

Yes. The data are available without restriction. I cannot comment on whether these are deposited in appropriate repositories as this information is not available.

Are details of the methodology sufficient to allow the experiments to be reproduced?

Yes.

Is the manuscript well organized and written clearly enough to be accessible to non-specialists?

Yes. However, certain key elements should be more clearly outlined to make it accessible to non-specialists such as the difference between bimodal and unilateral CI users.

Specific comments:

Line 65. “contralateral” implant – it is not clear what it is contralateral to (although I assume the first implant). Using “second implant” would be clearer for a naïve reader.

72. ...the secondary outcomes WERE disease...

74 ...were compared USING THE WILCOXEN SIGNED-RANK TEST....

74 statistical association between what? Outcomes and clinical characteristics of the CI user? Please specify.

84. 1-16 years of inter-implant delay is a very large range that includes children with minimal and short duration of delays as well as long delays. However, the authors classify it here as a “great inter-implant interval” with the main claim being that benefit can be obtained despite a great interval. However, it is possible that these benefits observed are being driven by those with minimal and short delays. This is difficult for the reader to assess given the absence of plots for data visualization.

97 – conditional reimbursement – please briefly explain what this means in this context

100 – what does “minor” hearing performance mean? Poorer speech perception/worse outcomes compared to short-delay or simultaneously implanted?

157 – if describing male or female then please refer to “sex” rather than “gender”, unless they were asked what they gender identify with

157 – age at time of speech test should also be reported for pre and post measures

176 – please outline how frequency of CI use was determined. Was this self-report or from datalogging built into the CI device?

184 – 186. Some children wore a unilateral CI only and so experienced unilateral deprivation/stimulation prior to receiving the second CI. Whereas others wore a unilateral CI with a contralateral hearing aid and so experienced bimodal input (i.e. bilateral access to sound from a combination of electric and acoustic input). These are two very different groups and this should be highlighted to the reader from the very beginning of the manuscript. Furthermore, literature on bimodal users should also be reviewed. The definition of inter-implant delay therefore differs for these groups and should be explicated. I.e. duration of unilateral deprivation/stimulation, vs duration of bimodal stimulation.

Table 4: “Gender, Men, Woman” should read “Sex, Male, Female”

References:

Easwar, V., Sanfilippo, J., Papsin, B., & Gordon, K. (2018). Impact of Consistency in Daily Device Use on Speech Perception Abilities in Children with Cochlear Implants: Datalogging Evidence. Journal of the American Academy of Audiology, 29(9), 835–846. https://doi.org/10.3766/jaaa.17051

Polonenko, M. J., Papsin, B. C., & Gordon, K. A. (2018). Limiting asymmetric hearing improves benefits of bilateral hearing in children using cochlear implants. Scientific Reports, 8(1), 13201. https://doi.org/10.1038/s41598-018-31546-8

Reviewer #2: The presented paper presents the results of a prospective study of a multi-centre study in sequential bilateral cochlear implantation in children between 5 to 18 years. This revised version is good to read, but there are still some things that are misleading for the uninitiated reader.

So far, I am still missing a clear research question as far as the main message is concerned. The paper contains a collection of speech comprehension data for the patient group. These have also been presented in dependence of predictors in other papers (e.g. Illg et al. 2019, which was also cited). Page 4, line 110 et seq. is described: "The speech perception outcome of

sequential BICI in older children and adolescents and the influence of preoperative residual hearing is not clear from the literature so far" What do you mean with this sentence? What is the point?

In the evaluation, the 10 non-users are listed separately to describe the group. However, their speech perception results are not included in the calculations. When conducting a prospective study, these data would also have to be calculated, or did these patients no longer appear at the 12-month deadline? It is not clear which result in speech understanding is achieved by a non-user, because there are different reasons for this. However, these data also belong to the total group and must not be eliminated, according to my understanding.

I also find it very difficult that patients with hypoplasia of the cochlea and the like were included in the total group, because they have different preconditions than children and adolescents with regular anatomy. From my point of view, these two groups should be separated and considered separately.

Furthermore, I do not understand in table 2 the CVC score in quiet (no HA) values for the bilateral CI condition and likewise the CVC scores in noise (no HA) in bilateral condition. Why do the have the same values like in daily life? If these are the results of a measurement without HA, then unilateral measurements must have been taken, or? Why is this listed under bilateral condition?

The description of tabel 1 (page 12 line 279 et seq.) is partly redundant because of adding the table.

Please correct the following sentencs page 18, line 400 "language skills might be eventually seen in de the ....."

My request for the further revision is that a clear question emerges on which the discussion can also be guided and that the data evaluation takes place in clearly defined groups.

6. PLOS authors have the option to publish the peer review history of their article (what does this mean?). If published, this will include your full peer review and any attached files.

Reviewer #1: No

Reviewer #2: No

---

## [Author Response · Author response to Decision Letter 0]

21 Jun 2021

Dear editor,

We would like to thank the editor for the opportunity to resubmit the manuscript entitled ‘Benefit of sequential bilateral cochlear implantation in children between between 5 to 18 years old: a prospective cohort study’. 

We submitted this paper last year and received valuable comments of the reviewers of PLOS ONE with the advice to revise the paper thoroughly before publication could be considered. Based on these comments we revised the paper carefully to improve the quality of the text including the statistical analysis and conscientiously revised the discussion. This resulted in an improved manuscript which we would like to offer to PLOS ONE for reconsideration of publication. Our detailed responses to the reviewers’ comments of the previous submission can be found in the attached file 'Response to Reviewers'. 

Thank you for considering our manuscript for publication. 

Sincerely,

On behalf of the author-team,

Drs. W.J. Kleijbergen

For journal use only: [PONE-D-20-37958] - [EMID:c008340100aa58f8]

---

## [Decision Letter · Decision Letter 1]

3 Nov 2021

PONE-D-20-37958R1Benefit of sequential bilateral cochlear implantation in children between 5 to 18 years old: a prospective cohort studyPLOS ONE

Dear Dr. Kleijbergen,

Thank you for submitting your manuscript to PLOS ONE. After careful consideration, we feel that it has merit but does not fully meet PLOS ONE’s publication criteria as it currently stands. Therefore, we invite you to submit a revised version of the manuscript that addresses the points raised during the review process.

We look forward to receiving your revised manuscript.

Kind regards,

Andreas Buechner

Academic Editor

PLOS ONE

Reviewers' comments:

Reviewer's Responses to Questions

**Comments to the Author**

1. If the authors have adequately addressed your comments raised in a previous round of review and you feel that this manuscript is now acceptable for publication, you may indicate that here to bypass the “Comments to the Author” section, enter your conflict of interest statement in the “Confidential to Editor” section, and submit your "Accept" recommendation.

Reviewer #2: All comments have been addressed

Reviewer #3: (No Response)

2. Is the manuscript technically sound, and do the data support the conclusions?

Reviewer #2: Yes

Reviewer #3: No

3. Has the statistical analysis been performed appropriately and rigorously? 

Reviewer #2: Yes

Reviewer #3: No

4. Have the authors made all data underlying the findings in their manuscript fully available?

Reviewer #2: Yes

Reviewer #3: No

5. Is the manuscript presented in an intelligible fashion and written in standard English?

Reviewer #2: Yes

Reviewer #3: No

6. Review Comments to the Author

Reviewer #2: Dear authors,

Thank you for the revision of your submission. For me, there are now still a few small things that should be revised:

Please revise the references in the text. In the chapter Discussion, for example, the references do not have associated numbers, so that one can then find them.

P. 13 line 357 The dot from the end of the sentence belongs after the bracket (Table 1).

The last two sentences in the Conlcusion section are slightly misleading, please adjust them. First you write that the inter-implant distance has no significant influence on the results and then you conclude in the last sentence that a larger inter-implant distance could have an influence.

Reviewer #3: The manuscript presents a multi-centre prospective study of the benefit of sequential cochlear implantation on speech, language, and quality of life outcomes. The study examines a group (n=85) or children (5 – 18 yrs old) who have experienced variable durations of delay between receiving their first and second implant.

This is a revised version of a previously revised manuscript. The authors have made improvements to the clarity of the research objectives and have improved the framing on these within the context of existing research. However, my previous suggestions for how to address some key problems and improve the manuscript have not been addressed, particularly regarding the level of transparency and technical standards of the methods and statistical analyses. I reiterate these recommendations below and include further comments that I hope the authors will find useful. The conclusions are not supported by the data and do not make a valid contribution to the field, nor enable parents or clinicians to make more informed decisions.

General comments:

1. The authors’ main claim is that children showed significant improvements in speech recognition in quiet and in noise despite a great inter-implant interval. However, this is not explored in the manuscript: changes in children’s outcomes over time are not reported (e.g. what percentage showed improvements, no improvement, or got worse?) and whether this level of improvement was impacted by inter-implant interval has not been explored. The statistical effects of inter-implant interval have not been reported for the two subgroups separately (‘bimodal’ children who wore a hearing aid during this inter-implant interval and ‘unilateral’ children who did not). If the level of benefit and predictive factors of the benefit of sequential implantation is to be truly understood, then the effect of inter-implant interval must be explored, and also reported for each group separately rather than collapsing the effect across the two groups and (potentially erroneously) claiming it is irrelevant to speech outcomes.

2. Developmental effects have not been examined or controlled for in analyses. The sample is heterogenous in terms of age and so confounding factors such as linguistic maturation over time must be reported. Likewise, the effect of age at CI1 on bilateral outcomes has not been reported. Again, this is highly variable in the sample, and is a key factor to consider when evaluating the level of benefit of sequential implantation and predictive factors.

3. This is a prospective cohort study that, at first look, appears to employ a longitudinal, repeated-measures design to compare pre- and post-CI2 outcomes to address the main objective of determining the benefit of sequential implantation. However, it is unclear for which participants repeated measurements were collected, or whether different groups were in fact assessed at each time point. This confusion is further compound by the author’s contradictory statement in line 264-265: ‘Between group differences were tested using Wilcoxon signed rank test (for related samples)’. Were two separate cohorts assessed at each time point, or were the same children assessed at pre- and post-CI2? The manuscript should be more transparent about what participants were tested and when, and therefore whether between-group or within-subject statistical analyses methods were employed.

4. Further to the above point, there appears to be much missing follow-up (and/or baseline) data. The authors state that missing data has been handled using multiple imputation. This could mean that missing data has been replaced with the mean or a predicted value. However, the authors provide no information about the methods involved and how the missing data was imputed. For repeated measures analyses, only participants with both baseline and follow-up data should be included.

5. The total number of participants and the number of data sets analysed is inconsistent throughout the manuscript and difficult to navigate. For example, Figure 1 reports that for CVC in silence post-CI2 n=63, but in Table 4 this is n=61 for the first 4 variables. Furthermore, in Table 4 reports that for CVC in silence post-CI2 use of hearing aid (yes/bimodal = 29, no/unilateral = 13) total n = 42. But in Table 2 CVC in silence outcomes use of hearing aid (bimodal = 33, unilateral = 30) total n = 63.

Specific comments:

1. Table 2 – Effects of sequential implantation on CVC speech perception (comparison of ‘bilateral CI’ to ‘CI1’) have been helpfully presented separately for bimodal and unilateral subgroups as well as together. Please do the same for the language outcomes presented in this table (even if the n is small for each subgroup, it is still useful for the reader to see this information).

2. Table 2 – Why are the 12-month unilateral CI2 outcomes not reported? Baseline Pre-CI2 is reported but post-CI2 are not, only bilateral outcomes are reported. It seems central to the main objective of the manuscript to understand if implantation of the second ear improved outcomes for this ear, especially for those who wore a hearing aid before implantation.

3. Table 2 – condition labels used in column headings are unclear and footnotes are confusing. For improved clarity, please consider using more descriptive labels such as ‘baseline CI1/best-aided’, ‘baseline CI2’, ‘postoperative bilateral’, (and also include ‘postoperative CI2’ as mentioned above) and remove footnotes where possible. These labels should be defined in ‘Outcome assessment’ (line 200) or similar.

4. Table 2 – CVC score in quiet ‘Baseline Pre-CI2’: Bimodal (n=13) and unilateral (n=17) gives a total n=30, not n=36 as reported here. What is the discrepancy in the reported numbers?

5. Table 4 – those with support of speech therapy (‘yes’) had lower outcomes than those without (‘no’). Does this mean that children received speech therapy because they were performing poorly and therefore needed it, whereas ‘good’ performers did not need speech therapy? Without context or interpretation, there is a risk that parents may see this result and think that speech therapy is detrimental to outcomes.

6. Table 4 – CELF, use of hearing aid, p value is reported as ‘>0.05’. Please report exact p value consistent with all other reporting of p values in this table.

7. I understand why separate regression analyses were conducted for each outcome measure (e.g. CVC, PPVT, CELF). However, for each of these separate outcomes, there are multiple predictors that are analysed separately. Therefore, all results reported in Table 4 have not been corrected for multiple comparisons and have not been estimated whilst controlling for covariates. This should be clearly stated.

8. Why were the effects of predictors on CVC scores in noise not analysed, especially given that this outcome was seen to improve post-CI2?

9. Line 383 – 387 No sig effect of inter-implant delay on CVC words across whole group, but was there an interaction? See next comment.

10. S1 Fig 2a/b – Please report the associated statistics (i.e., effect size, p value, and/or regression line equation) and n = for each regression line shown.

11. I am not familiar with the terms ‘Univariable’ and ‘Multivariable’. When talking about assessing the effect of a single predictor variable on a single outcome variable, the standard term is ‘univariate’ or ‘simple linear regression’. Likewise, the standard term for assessing the effect of several predictor variables on one outcome variable is ‘multivariate’ or ‘multiple linear regression’. Please clarify the analysis method used and use standard terminology throughout.

12. Line 387 ‘Multivariable regression analysis was not performed as no significant associations were tested in univariable regression analysis’. Please be aware that if a predictor is not significant in a univariate analysis, it could still be significant in a multivariate relationship once other effects have been incorporated in the model’s estimates. Consider rephrasing or expanding on why a multivariate regression was not performed on those factors with predictive effects (even if they did not reach significance threshold, e.g. moderate effects of age at CI2 and inter-implant delay).

13. SSQ ‘quality’ domain significantly improved from baseline to post-CI2. This is important and should be reported in the conclusion and abstract. Subsequently, please correct in abstract ‘No significant effect was seen on language outcomes and SSQ’ as the findings reported do not support this statement.

14. Line 430 – when discussing improvements in the SSQ ‘qualities’ domain, the authors state that ‘This outcome repletes the objective measurements of the speech perception scores in quiet and noise as descripted above’. In fact, the qualities domain most closely reflects/represents quality and effort of listening, rather than speech understanding ability. The ‘speech’ domain is the scale that most closely reflects speech understanding ability.

7. PLOS authors have the option to publish the peer review history of their article (what does this mean?). If published, this will include your full peer review and any attached files.

Reviewer #2: No

Reviewer #3: No

---

## [Author Response · Author response to Decision Letter 1]

26 Jan 2022

Dear editor and reviewers,

We first would like to thank the editor for the opportunity to resubmit the manuscript entitled ‘Benefit of sequential bilateral cochlear implantation in children between 5 to 18 years old: a prospective cohort study’. We would like to thank also the reviewers for the valuable comments, in this way we were encouraged to made a major revision. Based on these comments we revised the paper carefully to improve the quality of the text including the statistical analysis and reconstructed Table 2 and 4 in detail. This resulted in a strongly improved manuscript which we would like to offer to PLOS ONE for reconsideration of publication.

Thank you very much for considering our manuscript for publication. 

Sincerely,

On behalf of the author-team,

Drs. W.J. Kleijbergen

---

## [Decision Letter · Decision Letter 2]

4 Apr 2022

PONE-D-20-37958R2Benefit of sequential bilateral cochlear implantation in children between 5 to 18 years old: a prospective cohort studyPLOS ONE

Dear Dr. Kleijbergen,

Thank you for submitting your manuscript to PLOS ONE. After careful consideration, we feel that it has merit but does not fully meet PLOS ONE’s publication criteria as it currently stands. Therefore, we invite you to submit a revised version of the manuscript that addresses the points raised during the review process.

There is only one last small issue by one of the reviewers remaining. If you address that point quickly, I can assure immediate feedback and fast processing of your manuscript.

We look forward to receiving your revised manuscript.

Kind regards,

Andreas Buechner

Academic Editor

PLOS ONE

Journal Requirements:

Reviewers' comments:

Reviewer's Responses to Questions

**Comments to the Author**

1. If the authors have adequately addressed your comments raised in a previous round of review and you feel that this manuscript is now acceptable for publication, you may indicate that here to bypass the “Comments to the Author” section, enter your conflict of interest statement in the “Confidential to Editor” section, and submit your "Accept" recommendation.

Reviewer #2: All comments have been addressed

Reviewer #3: All comments have been addressed

2. Is the manuscript technically sound, and do the data support the conclusions?

Reviewer #2: Partly

Reviewer #3: Yes

3. Has the statistical analysis been performed appropriately and rigorously? 

Reviewer #2: Yes

Reviewer #3: Yes

4. Have the authors made all data underlying the findings in their manuscript fully available?

Reviewer #2: Yes

Reviewer #3: Yes

5. Is the manuscript presented in an intelligible fashion and written in standard English?

Reviewer #2: Yes

Reviewer #3: Yes

6. Review Comments to the Author

Reviewer #2: Thank you for the revision of the article: "Benefit of sequential bilateral cochlear implantation in children between 5 to 18 years old: a prospective cohorte study"

The data is now easier to follow, but I miss one basic blocks in the discussion. I miss where the methodological choice of the selected test procedures for the outcome are critically discussed in the light of other studies. For example, in your study, CVC was used for speech perception. Other research groups with a similar topic use monosyllable comprehension or tests with meaningful words for speech perception. It has to be discussed why you used CVC especially in the light of the fact that no significant correlations with implant age or inter-implant interval were found. Speech perception with CVC does not mean speech understanding of words.

Likewise, a fundamental discussion of the critical time period of speech development based on scientific principles (e.g. Sharma or Kral) is missing.

It would be good if they put a paragraph like that in the discussion.

Reviewer #3: The authors have made significant improvements to the manuscript and have adequately addressed my previous concerns. I am happy to see this important research published and commend the authors on their efforts.

I only have two minor comments that I would encourage the authors to consider:

1. When referring to frequency of CI2 use on page 14 and in Table 1&4, please change ‘regularly use’ to ‘frequent use’ to keep the subdivision naming consistent with definitions on page 7 line 197.

2. Please provide Wilcoxon signed ranks Z test statistic in the results tables and/or in the manuscript text. This will enable the reader and any future systematic reviews to calculate effect sizes.

7. PLOS authors have the option to publish the peer review history of their article (what does this mean?). If published, this will include your full peer review and any attached files.

Reviewer #2: No

Reviewer #3: No

---

## [Author Response · Author response to Decision Letter 2]

25 Apr 2022

Dear editor,

We would like to thank you again for the opportunity to resubmit the manuscript entitled ‘Benefit of sequential bilateral cochlear implantation in children between 5 to 18 years old: a prospective cohort study’. We added or changed the specific parts, mentioned in the valuable comments of the reviewers and re-submitted our paper again. Our detailed responses to the reviewers’ comments of the previous submission can be found in the document 'Response to Reviewers'.

Dear reviewer 2,

We would like to thank the reviewer for the valuable comment to explain the methodological choice of the selected test procedures for the outcome of speech perception and secondly for the encouragement to add an additional part to the discussion about the critical time of period of speech perception. 

Firstly, we want to underline that we tested speech perception with meaningful words. These words were monosyllabic and had a CVC-structure. In the Netherlands, this is a standardized test that is used for measuring speech recognition. We feel that the reason that the outcomes were not correlated with age of the second cochlear implantation or inter-implant interval was rather related to the time of onset of severe hearing loss rather than the use of our speech recognition task. To clarify this to the reader that these CVC words were meaningful, we added this to methods section.

Secondly, we added a part to the discussion were we discuss the fundamental issue of the critical time period of the inter-implant delay and limited device use on auditory brainstem and subsequent cortical maturation to explain our results in more detail. 

Dear Reviewer 3, 

We would like to thank the reviewer for the feedback. We changed the text accordingly. We also added the Wilcoxon signed ranks Z test statistic in Table 2 and 3 accordingly. 

Sincerely,

On behalf of the author-team,

Drs. W.J. Kleijbergen

---

## [Decision Letter · Decision Letter 3]

5 Jul 2022

Benefit of sequential bilateral cochlear implantation in children between 5 to 18 years old: a prospective cohort study

PONE-D-20-37958R3

Dear Dr. Kleijbergen,

We’re pleased to inform you that your manuscript has been judged scientifically suitable for publication and will be formally accepted for publication once it meets all outstanding technical requirements.

Kind regards,

Andreas Buechner

Academic Editor

PLOS ONE

Additional Editor Comments (optional):

Reviewers' comments:

Reviewer's Responses to Questions

**Comments to the Author**

1. If the authors have adequately addressed your comments raised in a previous round of review and you feel that this manuscript is now acceptable for publication, you may indicate that here to bypass the “Comments to the Author” section, enter your conflict of interest statement in the “Confidential to Editor” section, and submit your "Accept" recommendation.

Reviewer #2: All comments have been addressed

Reviewer #3: All comments have been addressed

2. Is the manuscript technically sound, and do the data support the conclusions?

Reviewer #2: Yes

Reviewer #3: Yes

3. Has the statistical analysis been performed appropriately and rigorously? 

Reviewer #2: Yes

Reviewer #3: Yes

4. Have the authors made all data underlying the findings in their manuscript fully available?

Reviewer #2: Yes

Reviewer #3: Yes

5. Is the manuscript presented in an intelligible fashion and written in standard English?

Reviewer #2: Yes

Reviewer #3: Yes

6. Review Comments to the Author

Reviewer #2: Good work! Thank you for revising and resubmitting the manuscript. By adding the new statistical values and revising the discussion, the article has gained significantly for the reader.

Reviewer #3: The authors addressed all my comments adequately and I can fully recommend the manuscript for publication in your journal.

7. PLOS authors have the option to publish the peer review history of their article (what does this mean?). If published, this will include your full peer review and any attached files.

Reviewer #2: No

Reviewer #3: No

---

## [Editor Report · Acceptance letter]

8 Jul 2022

PONE-D-20-37958R3 

Benefit of sequential bilateral cochlear implantation in children between 5 to 18 years old: a prospective cohort study 

Dear Dr. Kleijbergen:

I'm pleased to inform you that your manuscript has been deemed suitable for publication in PLOS ONE. Congratulations! Your manuscript is now with our production department. 

Kind regards, 

on behalf of

Andreas Buechner 

Academic Editor

PLOS ONE